# Impact of the COVID-19 Pandemic on the Mental Health of Nurses and Auxiliary Nursing Care Technicians—A Voluntary Online Survey

**DOI:** 10.3390/ijerph18168310

**Published:** 2021-08-05

**Authors:** Eduardo Sánchez-Sánchez, J. Ángel García-Álvarez, Esperanza García-Marín, María Gutierrez-Serrano, Maria José M. Alférez, Guillermo Ramirez-Vargas

**Affiliations:** 1Internal Medicine Department, Punta de Europa Hospital, 11207 Algeciras, Spain; juangelgarcia@hotmail.com (J.Á.G.-Á.); espe_garciamarin@hotmail.com (E.G.-M.); mariaguse@hotmail.com (M.G.-S.); guiram1992@gmail.com (G.R.-V.); 2Instituto de Investigación e Innovación Biomédica de Cádiz (INiBICA), Hospital Universitario Puerta del Mar, Universidad de Cádiz, 11009 Cádiz, Spain; 3Department of Physiology, Faculty of Pharmacy, Campus Universitario de Cartuja, University of Granada, 18011 Granada, Spain; malferez@ugr.es

**Keywords:** COVID-19, health workers, nurse, anxiety, depression, HADS, ANCTs

## Abstract

Pandemics impose an immense psychological burden on healthcare workers due to a combination of workplace stressors and personal fears. Nurses and auxiliary nursing care technicians (ANCTs) are on the front line of this pandemic and form the largest group in healthcare practice. The aim of this study is to determine the symptoms of depression and/or anxiety among nurses and ANCTs during the periods known as the first wave (March–June) and second wave (September–November) of theCOVID-19 pandemic in Spain. An observational cross-sectional study was carried out using an anonymous, self-administered questionnaire among nurses and ANCTs practising in Spain. During the first period, 68.3% and 49.6% of the subjects presented anxiety and depression, respectively, decreasing in the second period (49.5% for anxiety and 35.1% for depression). There were statistically significant differences between the different categories and periods (*p* < 0.001). The COVID-19 pandemic has negatively influenced mental health in nurses and ANCTs. Mental health should be monitored and coping strategies promoted to improve the health, productivity and efficiency of these professionals.

## 1. Introduction

In December 2019, cases of atypical pneumonia were reported in Wuhan, China, caused by a new virus with rapid spread, infectivity and mortality in humans. The World Health Organization (WHO) subsequently temporarily named this new virus as the new coronavirus 2019 (SARS-CoV-2), and the disease that it caused as COVID-19. The WHO classified this situation as a global public health emergency [1,2,3,4].

The COVID-19 pandemic is putting unprecedented pressure on health and social systems. As a result, most national governments took measures to reduce infections among the population. These measures have a negative impact on the economy of these countries [5,6].These measures and their negative repercussions may lead to an increase in mental disorders in the population, and in frontline workers such as health professionals [7].

In Spain, from March to June (“1st wave”), a state of alarm and a period of total confinement were imposed, followed by a de-escalation to allow the population to return to normality after this period [8]. During this de-escalation, some rules were relaxed, such as the opening of shops, the non-compulsory use of masks, the increase in catering capacity, movement between different geographical areas, etc. This relaxation led to an increase in COVID-19 cases between September and November (“2nd wave”), and new strict measures had to be taken to reduce the incidence.

Healthcare professionals were considered an essential group for the management of this pandemic. These professionals are a high-risk population group, as they are directly involved in the care of patients with COVID-19 [9]. The risk of infection in these professionals was high during the first wave, with the number of healthcare professionals infected with COVID-19 in Spain being 50,000 (21% of the total number infected in that period), compared to 30,000 in France and Italy and 15,000 in Germany [10,11]. Moreover, this figure increased to 118,063 positive healthcare workers on 28 January 2021, so that 4.3% of the total infected population was healthcare workers [12].

Pandemics impose an immense psychological burden on healthcare workers due to a combination of workplace stressors and personal fears [13,14], including lack of personal protective equipment (PPE) and other resources, long working hours, exposure to COVID-19-positive patients, fear of spreading the virus to family and relatives, increased workload, isolation, stigmatisation and the loss of a colleague, among others [15,16]. These factors contribute to increased physical and psychological burden on healthcare professionals [17], which can lead to the development of stress, insomnia, depression and/or anxiety [18].

Among healthcare professionals, nurses and auxiliary nursing care technicians (ANCTs) are at the frontline of this pandemic [19] and form the largest group in healthcare practice [20,21]. Moreover, nurses and ANCTs maintain a close relationship with their patients throughout hospital admission [22]. The loneliness of these patients, due to imposed isolation, has led to nurses and ANCTs accompanying these patients through very difficult times, including death, as their only support. This can affect their mood and mental health. This, together with the factors mentioned above, increases the risk that nurses and ANCTs may experience depression and/or anxiety, as well as post-traumatic stress related to the pandemic [19].

An increase in depression and/or anxiety in this group of healthcare professionals has been associated with a decrease in patient safety, which would negatively influence on healthcare [23]. Hence, studying the effect of the pandemic on the mental health of these professionals and how this may be modified over time to improve information and training is important. For this reason, anxiety and depression should be studied in various periods and the changes that have occurred in these periods and how they have affected health professionals.

The aim of this study is to determine the symptoms of depression and/or anxiety among nurses and ANCTs during the periods known as the “1st wave” (March–June) and “2ndwave”(September–November)of the COVID-19 pandemic in Spain, as well as the differences between the two periods.

## 2. Materials and Methods

### 2.1. Study Design and Participants

An observational cross-sectional study was carried out based on an online questionnaire using a web platform, where a non-probability sampling method, convenience or snowball sampling, was used, where the participants themselves disseminate and recruit other participants. Participants had to be nurses or ANCTs practising in Spain (inclusion criterion).

According to data provided by the Ministry of Health in 2020, there were 186,000 nurses in Spain [24]. There are no specific data on the number of ANCTs, but it has been estimated that in 2019 there were a total of 120,110 [25]. Therefore, a total of 306,110 nurses were working in Spain.

### 2.2. Instruments and Variables

The questionnaire was divided into three sections. In the first section, data were collected on socio-demographic variables, such as age and gender, and other variables such as ownership of the workplace management and work experience.

In the second part, variables related to the pandemic and emotional exhaustion were collected, such as: if they were isolated or a family member was isolated, if they felt fear due to lack of material, human or security resources, if they felt rejected, had sleep disturbances or consumed substances to relax.

The third section measured the degree of emotional distress (anxiety and depression) using the Hospital Anxiety and Depression Scale (HADS) [26] The HADS assesses symptom severity and cases of anxiety and depressive disorders in somatic, psychiatric and primary care patients and in the general population [27]. The scale was validated for the Spanish adult and healthy population, not only forhospitalised or ill patients [28,29]. It includes 14 items divided into two subscales (anxiety and depression), each with 7 items. The items are scored on a Likert type scale (0, 1, 2 and 3). The total scale score ranges from 0 to 21. The cut-off points for the different categories in the two subscales are: normal score (≤7), risk (8–10) and anxiety/depression (≥11) [13].

Section 2 and Section 3 were duplicated, i.e., they were collected for the “1st wave” and for the “2nd wave”.

### 2.3. Data Collection

The questionnaire was administered online on a free platform (Google) [30]. Social networks such as WhatsApp, Twitter, Facebook and Instagram were used for dissemination. A mass dissemination was carried out with the help of professional associations and scientific societies, among others. Responses were collected from 5 December 2020 to 5 February 2021. Completion of the questionnaire was voluntary.

### 2.4. Statistical Analysis

The results obtained were analysed separately, for each period, and together they were compared and related to the objective of this study.

The data obtained for the different variables were represented descriptively, by frequency and percentage, and the quantitative variables were expressed by the mean and standard deviation or dispersion. Subsequently, using Pearson’ schi-square test, significant differences between different groups were evaluated. A confidence level of 95% was assumed. Significance was set at an alpha level of 0.05.

For the comparative study of the HADS results in those subjects who had worked in both periods, the McNemar–Browker test was performed, with an alpha level of 0.05.

Statistical processing was carried out with SPSS-25 software (IBM Corporation, Armonk, NY, USA).

## 3. Results

A total of 687 responses were obtained, but seven were eliminated because of missing data for the variables sex and age. Of the 680 remaining subjects, 25 were eliminated because there were discrepancies between the answers to some questions, for example, not working in a period, but answering affirmatively to the questions related to that period. After eliminating these subjects, the sample consisted of 655 subjects, of whom 627 subjects had worked in the first wave and 655 subjects in the second wave. The number of subjects who had worked in both periods was 627.

### 3.1. Socio-Demographic and Professional Variables

Table 1 shows the socio-demographic characteristics of both groups in each study period.

### 3.2. Pandemic-Related Variables and Emotional Exhaustion

The results showed that the number of respondents who were isolated increased slightly in the second period (23.8% vs. 24.8%), as well as the figures for family members with COVID-19 infection (24.1% vs. 31.6%).

During the first period, 40.0% of the subjects responded that they had felt fear due to lack of personal protective equipment (PPE) to carry out their work, followed by a lack of safety of their family related to their exposure at work (25.3%) and a lack of protocols (14.8%). These percentages changed in the second period, with lack of safety of the family being the most prevalent with 40.6%, followed by lack of human resources with 24.4%. Lack of PPE decreased to 4.0%.

Of the subjects,29.8% responded that they felt rejected during the first period, decreasing to 26.7% in the second period, with no statistically significant differences in both periods (*p =* 0.983).

At the beginning of the pandemic (first period), 28.4% of the subjects reported having sleep disturbances every day, decreasing to 14.3% in the second period. Before the pandemic, 84.1% did not take sleep aids, 6.2% took pharmacological substances, 8.1% took natural substances and 1.6% took both. When asked whether they took substances to improve relaxation, 10.2% took both type of substance in the first period, although this decreased to 9.5% in the second period. More natural than pharmacological substances were used in both periods (Table 2).

### 3.3. HADS, Anxiety Scale

Some of the subjects in the sample worked in both periods under study or only in one period. Six hundred and twenty-sevensubjects worked in the first period and 655 worked in the second period.

The data showed that during the first period, 68.3% of the subjects had anxiety and 18.2% were at risk of anxiety, with a mean score of 12.91 on the HADS. This score decreased to 10.60 in the second period, decreasing the percentage of subjects with anxiety to 49.5%, and increasing the number of subjects at risk (22.6%). All scores decreased in the second period in relation to the first period. The item with the highest score in both periods (2.26 and 1.79), was the one related to being nervous or tense. There were no statistically significant differences between the different items and study periods (Table 3).

### 3.4. Depression Scale HADS

The results showed that during the first period, 49.6% of the subjects were depressed and 21.4% were at risk of depression, with a mean HADS score of 10.66. This score decreased to 8.87 in the second period, decreasing the percentage of subjects with depression to 35.1%, with a slight increase in subjects at risk (23.3%). 

All scores decreased in the second period in relation to the first period, with the greatest decrease in the item related to loss of interest in the personal aspect (1.79 vs. 1.37). There were no statistically significant differences between the different items and study periods (Table 4).

### 3.5. Comparison of the HADS in Both Periods

A total of 627 subjects worked in the two periods under study.

The results showed that there were statistically significant differences (*p* < 0.001) between the different categories in the anxiety scale in the first and second periods (normal = 13.6% vs. 22.2%; risk = 18.2% vs. 28.1%; anxiety = 68.3% vs. 49.8%). Within the depression scale, the percentages of depression decreased (49.6% vs. 35.1%), with increasing percentages for the normal (29.0% vs. 41.0%) and risk (21.4% vs. 23.9%) categories.

The results showed that in the first period, the percentages of anxiety and depression were lower in nurses than in the ANCT group (anxiety = 76.0% vs. 64.7%; depression = 56.6% vs. 45.0%, respectively).

In the second period, frequencies decreased in both, with the difference for anxiety decreasing (nurses = 48.5% and ANCTs = 52.5%). The figures for depression decreased, but this decrease was smaller in the nurses’ group, with a final frequency of 36.2%, compared to 34.2% in the group of ANCTs.

There were significant differences in the two groups in the first and second periods (Table 5).

### 3.6. Odds Ratio Anxiety/Depression and Variables Recorded

In the first wave, women were less likely than men to have anxiety (OR = 0.59, CI: 0.35–0.98, *p =* 0.042), as were the group of ANCTs (OR = 0.59, CI: 0.37–0.92, *p =* 0.024) and those who had been isolated (OR = 0.58, CI: 0.36–0.90, *p =* 0.019) or had felt rejected (OR = 0.45, CI: 0.29–0.68, *p* < 0.001). In the second wave, the ANCT group (OR = 0.60, CI: 0.40–0.88, *p* = 0.010) and the subjects who felt rejected (OR = 0.35, CI: 0.24–0.51, *p <* 0.001) were less likely to have anxiety.

Women (OR = 0.37, CI: 0.21–0.63, *p <* 0.001), the ANCT group (OR = 0.55, CI: 0.36–0.83, *p =* 0.004), those who had an isolated family member (OR = 0.58, CI: 0.28–0.86, *p =* 0.008) and felt rejected (OR = 0.38, CI: 0.26–0.55, *p <* 0.001) were less likely to experience depression during the first wave. Those with 0–5 years or 6–10 years of experience were 2.46 (CI: 1.20–5.11, *p =* 0.014) and 2.32 (CI: 1.17–4.64, *p =* 0.015) times more likely to experience depression in the first wave, respectively. In the second wave, the ANCT group and the subjects who felt rejected were less likely to be depressed (Table 6).

## 4. Discussion

Since the onset of the pandemic, healthcare professionals have been subjected to a high emotional burden, which has negatively influenced their mental health, and healthcare professionals, nurses and ANCTs play a key role in the treatment of infectious diseases [31,32]. During the COVID-19 pandemic, nurses and ANCTs have suffered a high physical and psychological burden [19], as they have longer and closer contact with COVID-19 patients [33], and are therefore at higher risk of infection than other healthcare professionals.

Our research has studied the presence of anxiety and depression among nurses and ANCTs, as well as other stressors during that first period and a second period when the incidence increased again. The results show that anxiety and depression were more prevalent in the first period (68.3% and 49.6%), decreasing in the second period understudy (49.5% and 35.1%). This decrease in the second period may be due to the fact that at the beginning of the pandemic, the “first wave”, there was a lot of confusion, lack of information, lack of training of these professionals, lack of PPE, large number of infections in both groups and a lack of diagnostic tests. In the second period, the training of these professionals improved, as well as the material and human resources for the management of this pandemic. Most research has studied the incidence of anxiety and depression among healthcare professionals, either during the period of confinement or immediately after confinement. Hummel et al. conducted a study among healthcare professionals in eight European countries and their results showed that 37% of the surveyed professionals had moderate or severe anxiety and 35% had moderate or severe depression [34]. In a study conducted in China, 44.7% of health professionals were anxious and 55.7% were depressed [35]. In Turkey, 36.9% and 57.5% of health professionals experienced anxiety and depression, respectively [36]. In Spain, after the first wave of the pandemic (April–May) 79.5% and 51.1% of health professionals experienced anxiety and depression [37]. Another national investigation, in a geographical area that experienced a high incidence of COVID-19 (Igualada, Catalunya, Spain), showed that 71.6% of health professionals surveyed experienced anxiety and 60.3% depression [38]. These differences in the incidence of anxiety and depression may be due to multiple influencing factors. These factors include: the measurement instrument used, the country where the study was conducted, the incidence of COVID-19 in that country, the type of healthcare system, the pressure of care in healthcare facilities, the time period of data collection and the study population. The spread of the pandemic in each territory has determined the magnitude of the emotional response [9]. In addition, most studies have been conducted in physicians or healthcare professionals in general, and our research has been conducted in nurses and ANCTs, who report a higher psychological burden [39].Furthermore, the scores obtained in the HADS by other authors are lower than those obtained in our study, with the anxiety scale value in our study being 12.91 in the first period and 10.60 in the second period, and adepression scale value of 10.66 and 8.87 in the first and second periods, higher than those reported by Tan et al. (6.9 and 5.7, respectively) [13].

Our results show that women were less likely to be anxious than men, as well as ANCTs versus nurses. These results are contrary to those reported by other authors, who concluded that ANCTs were more anxious than nurses [40], as well as the female gender [41,42]. There are more women in these professions, and this may explain the results obtained. The roles of nurses and ANCTs are different, especially those related to basic patient care, which may influence their exposure time and distance from patients, decreasing or increasing the risk of infection. Nurses and ANCTs with less experience (0–5 and 6–10 years) were more likely to be depressed. There are no references in relation to increased depression in one group or the other, but there are references in relation to increased stress in less experienced professionals [16]. This may be due to the choice of data collection questionnaire, timing of data collection and different geographical areas. The impact and management of the pandemic is different depending on the country or geographical area.

Other aspects that may influence the mental health of nurses and ANCTs are the response of governments and health policies and the capacity to acquire human and material resources. The lack of PPE, increased loneliness and deaths in these patients [43], together with the risk of self and/or family contagion [44,45,46], have played an important role in the occurrence of anxiety and/or depression among nurses and ANCTs. Our results showed that in the first period, lack of PPE and security of the family were the factors with the highest psychological burden. During the first period of the pandemic, the shortage of PPE gave rise to fear and changes in protocols [31,45], which may explain the results obtained in our study, as nurses and ANCTs, being unsure due to the lack of PPE, reported greater fear of infecting their family, as the number of infections of family members increased in the second wave. Our results (25.3% and 40.6%) were lower than those reported in other research (75.2%) [47]. In the second period, there was no shortage of PPE, so the percentage decreased significantly (4.0%). The lack of human resources and the fear of infecting the family increased significantly. This may be due to an increase in cases in the second wave, associated with the measures taken by government officials being less strict than the initial ones (lockdown) and the continuous sick leave due to increased preventive isolations of nurses and ANCTs, due to close contacts with patients or other COVID-19-positive professionals.

During the pandemic, the rejection of part by the population of health professionals became visible. Almost three out of 10 nurses and ANCTs felt rejected, which can affect their mental health.

Psychological burden, anxiety and/or depression during a pandemic may affect the sleep of healthcare professionals, increasing insomnia or the frequency of sleep interruptions [48,49]. Of our study sample, 28.4% of respondents reported sleep disturbance every day during the first wave, decreasing to half (14.3%) during the second wave. These data are lower than those reported by other authors [35,42]. Only 8.1% of the respondents answered that they never had sleep disturbances during the first period, increasing to 11.1% during the second period. This may be due to the decrease in fear due to the lack of PPE, improved protocols, etc. The use of pharmacological and/or natural substances to sleep or relax also decreased in the second period (42.7%) compared to the first period (47.9%). The use of natural substances was higher than the use of pharmacological substances.

Among the limitations of the study, the type of sampling used in this study may give rise to a selection bias, as the sample was obtained in a non-randomised manner. This bias has been assumed because access to the study population at the national level was very difficult due to the situation in which the Spanish health system was immersed and the mobility restrictions in that period. The low representation of men in relation to the population could lead to gender bias. It should be remembered that the female sex is the most prevalent sex in these professions (nurses and ANCTs). In addition, there is a higher representation of nurses than of ANCTs. Another limitation present is the difference in representation by age, so the results should be interpreted with caution. In addition, the use of an online questionnaire may generate acquiescence response bias, although the questionnaire was simplified to reduce response time. However, Ekman et al., in 2006, stated that the bias with the collection of information via web questionnaires was no greater than that caused by paper questionnaires [50].

The general is ability of the results is limited, as it is a non-probability sample and due to the biases mentioned above. Moreover, as it is a voluntary questionnaire, those nurses and/or ANCTs who are more aware of the problem or who have experienced a greater impact on their emotional state may have participated. Another limitation is the possibility of bias when subjects recall or interpret their past relative to their current emotional states, but this bias may be common in questionnaires of this type due to subjects’ perceptions and memories of their emotional state at the time. In addition, the context or climate may influence the psychological health and responses of respondents. However, it should be noted that, although the current research could not be representative of the entire population of nurses and/or ANCTs working in Spain, a broad coverage has been achieved, being a starting point for more specific future research.

## 5. Conclusions

The COVID-19 pandemic has had a negative impact on the mental health of healthcare professionals, especially nurses and ANCTs, due to the increased burden of care or the fear of facing a new or highly stressful situation.

Symptoms of anxiety and/or depression have been present in these professionals during the pandemic, being greater in the first period than in the second, due to a lack of knowledge, lack of resources and uncertainty at the beginning of the pandemic. These symptoms can have a negative impact on healthcare and clinical safety in the population, which could increase the number of deaths in this pandemic.

Therefore, mental health should be monitored and coping strategies should be promoted among nurses and ANWs, with the aim of improving the health, productivity and efficiency of these professionals. In addition, public policies should be created to address mental health in these professionals, which can anticipate solutions to situations similar to the one experienced during the COVID-19 pandemic.

## Figures and Tables

**Table 1 ijerph-18-08310-t001:** Socio-demographic and occupational variables.

	1st Wave (*n* = 627) (*n*, %)			2nd Wave (*n* = 655) (*n*, %)		
	Nurses	ANCTs	*p*-Value	Nurses	ANCTs	*p*-Value
Gender: • Female • Male			<0.001 ****			<0.001 ****
	361 (83.8)	184 (93.9)		382 (83.6)	187 (94.4)	
	70 (16.2)	12 (6.1)		75 (16.4)	11 (5.6)	
Age: • 16–25 years • 26–35 years • 36–45 years • 46–55 years • ≥56 years			<0.001 ****			<0.001 ****
	49 (11.4)	6 (3.1)		59 (12.9)	7 (3.5)	
	135 (31.3)	37 (18.9)		149 (32.6)	38 (19.2)	
	140 (32.5)	68 (34.7)		145 (31.7)	67 (33.8)	
	88 (20.4)	61 (31.1)		87 (19.0)	64 (32.3)	
	19 (4.4)	24 (12.2)		17 (3.7)	22 (11.1)	
Institution: • Public • Private • Concerted • Public and private * • Public and concerted * • Private and concerted *			<0.001 ****			0.001 ****
	358 (83.1)	142 (72.4)		377 (82.5)	141 (71.2)	
	22 (5.1)	28 (14.3)		22 (4.8)	30 (15.2)	
	24 (5.6)	18 (9.2)		25 (5.5)	18 (9.1)	
	21 (4.9)	5 (2.6)		24 (5.3)	6 (3.0)	
	3 (0.7)	1 (0.5)		6 (1.3)	1 (0.5)	
	3 (0.7)	2 (1.0)		3 (0.7)	2 (1.0)	
Work experience: • 0–5 years • 6–10 years • 11–20 years • >20 years			0.187			0.308
	100 (23.2)	56 (28.6)		123 (26.9)	59 (29.8)	
	77 (17.9)	32 (16.3)		81 (17.7)	34 (17.2)	
	131 (30.4)	66 (33.7)		131 (28.7)	65 (32.8)	
	123 (28.5)	42 (21.4)		122 (26.7)	40 (20.2)	

ANCTs: auxiliary nursing care technicians. * Subjects working in two institutions simultaneously; ****< 0.001.

**Table 2 ijerph-18-08310-t002:** Pandemic-related variables and emotional exhaustion.

	1st Wave (*n*= 627)		2nd Wave (*n* = 655)		1st and 2nd Wave	
	*n* (%)	*p*-Value	*n* (%)	*p*-Value	X^2^	*p*-Value
Isolation by COVID-19: • Yes • No		<0.001 ***		<0.001 ***	1.10	0.293
	149 (23.8)		162 (24.8)			
	478 (76.2)		493 (75.1)			
A family member was infected by COVID-19: • Yes • No		<0.001 ***		<0.001 ***	0.10	0.754
	151 (24.1)		207 (31.6)			
	476 (75.9)		448 (68.4)			
Fear for lack of: • PPE • Protocols • Human resources • Own security • Security offamily • Diagnostic tests		<0.001 ***		<0.001 ***	18.98	0.798
	251 (40.0)		26 (4.0)			
	93 (14.8)		76 (11.6)			
	33 (5.2)		160 (24.4)			
	56 (8.9)		67 (10.2)			
	159 (25.3)		266 (40.6)			
	35 (5.6)		60 (9.2)			
Feelings of rejection: • Yes • No		<0.001 ***		<0.001 ***	0.00	0.983
	187 (29.8)		175 (26.7)			
	440 (70.2)		480 (73.3)			
Sleep disturbances: • Never • Perhaps • Sometimes • Many times • Every day		<0.001 ***		<0.001***	17.91	0.928
	51 (8.1)		73 (11.1)			
	15 (2.4)		13 (2.0)			
	140 (22.3)		246 (37.5)			
	243 (38.8)		229 (34.9)			
	178 (28.4)		94 (14.3)			
Substances to enhance relaxation: • Yes, pharmacological substances • Yes, natural substances • Both • No		<0.001 ***		<0.001 ***	8.10	0.524
	113 (18.0)		99 (15.1)			
	123 (19.6)		118 (18.0)			
	64 (10.2)		62 (9.5)			
	327 (52.1)		376 (57.4)			

PPE: personal protective equipment; X^2^: Pearson’s chi-square; ***< 0.001.

**Table 3 ijerph-18-08310-t003:** HADS, anxiety scale.

Items HADS Anxiety (Score)	1st Wave (*n*= 627) (*n*,%)	X	SD	2nd Wave (*n* = 655) (*n*,%)	X	SD	X^2^	*p*
Have you ever felt tense or nervous? • Not at all (0) • From time to time, occasionally (1) • A lot of the time(2) • Most of the time (3)		2.26	0.77		1.79	0.79	11.75	0.466
	8 (1.3)			17 (2.6)				
	102 (16.2)			234 (35.7)				
	235 (37.5)			269 (41.1)				
	282 (45.0)			17 (2.6)				
Have you ever felt a sense of fear as if something horrible was going to happen to you? • Not at all (0) • A little, but it does not worry me (1) • Yes, but not too badly (2) • Very definitely and quite badly (3)		1.92	0.97		1.64	0.94	19.32	0.081

	62 (9.9)			81 (12.4)				
	138 (22.0)			208 (31.7)				
	209 (33.3)			227 (34.6)				
	218 (34.7)			139 (21.2)				
Was your mind full of worries? • Only occasionally (0) • From time to time, but not too often (1) • A lot of the time (2) • A great deal of the time (3)		2.07	0.97		1.67	0.95	9.29	0.677
	65 (10.4)			99 (15.1)				
	82 (13.1)			142 (21.7)				
	220 (35.1)			285 (43.5)				
	260 (41.5)			129 (19.7)				
Did you have strange, “fluttering” feeling in your stomach? • Not at all (0) • Occasionally (1) • Quite often (2) • Very often (3)		1.54	0.99		1.25	0.91	10.28	0.591

	97 (15.5)			135 (20.6)				
	223 (35.6)			293 (44.7)				
	174 (27.7)			153 (23.3)				
	133 (21.2)			74 (11.3)				
Did you feel restless, as if you couldn’t stop moving? • Not at all (0) • Not very much (1) • Quite a lot (2) • Very much indeed (3)		1.64	0.92		1.42	0.86	8.65	0.732

	75 (11.9)			94 (14.3)				
	195 (31.1)			260 (36.7)				
	235 (37.5)			231 (35.3)				
	122 (19.4)			70 (10.7)				
Did you have sudden feelings of panic? • Not at all (0) • Not very often (1) • Quite often (2) • Very often indeed (3)		1.51	1.06		1.20	0.94	20.08	0.066
	129 (20.6)			170 (25.9)				
	194 (30.9)			250 (38.2)				
	155 (24.7)			167 (25.5)				
	149 (23.7)			68 (10.4)				
Could you sit quietly and feel relaxed? • Definitely (0) • Usually (1) • Not very often (2) • Not at all (3)		1.93	0.75		1.60	0.70	9.11	0.693
	18 (2.8)			24 (3.6)				
	144 (23.0)			273 (41.7)				
	323 (51.5)			299 (45.6)				
	142 (22.6)			59 (9.0)				
Category anxiety scale: • Normal (≤7) • Risk (8–10) • Anxiety (≥11)		12.91	5.01		10.60	4.69	6.12	0.409
	85 (13.5)			183 (27.9)				
	114 (18.2)			148 (22.6)				
	428 (68.3)			324 (49.5)				

X: mean; SD: standard deviation; X^2^ Pearson’schi-square; *p*: *p*-value.

**Table 4 ijerph-18-08310-t004:** HADS, depression scale.

Items HADS Depression (Score)	1st Wave (*n*= 627) (*n*, %)	X	SD	2nd Wave (*n* = 655) (*n*, %)	X	SD	X^2^	*p*
Did you still enjoy what you used to like? • Definitely as much (0) • Not quite so much (1) • Only a little (2) • Hardly at all (3)		1.53	1.01		1.14	0.93	19.11	0.086
	109 (17.4)			106 (16.2)				
	210 (33.5)			258 (39.4)				
	171 (27.3)			193 (29.4)				
	135 (21.8)			98 (14.9)				
Could you laugh and see the funny side of things? • As much as I always could (0) • Not quite so much now (1) • Definitely not so much now (2) • Not at all (3)		1.30	0.94		1.02	0.78	12.69	0.890
	137 (21.8)			163 (24.9)				
	237 (37.8)			340 (51.9)				
	178 (28.4)			122 (18.6)				
	75 (11.9)			29 (4.4)				
Were you feeling cheerful? • Most of the time (0) • Sometimes (1) • Not often (2) • Not at all (3)		1.64	0.88		1.24	0.79	10.43	0.578
	63 (10.0)			109 (16.6)				
	205 (32.7)			315 (48.1)				
	251 (40.0)			196 (29.9)				
	108 (17.2)			35 (5.3)				
Did you feel like you were getting slower every day? • Not at all (0) • Sometimes (1) • Very often (2) • Nearly all the time (3)		1.32	0.92		1.19	0.85	20.29	0.062
	130 (20.7)			141 (21.5)				
	237 (37.8)			292 (44.6)				
	188 (30.0)			176 (26.9)				
	72 (11.5)			46 (7.0)				
Had you lost interest in your personal appearance? • I take just as much care as ever (0) • I may not take quite as much care (1) • I do not take as much care as I should (2) • Hardly at all (3)		1.79	1.10		1.37	0.99	6.33	0.898
	115 (18.3)			156 (23.8)				
	114 (8.2)			187 (28.5)				
	184 (29.3)			224 (34.2)				
	214 (34.1)			88 (13.4)				
Did you feel optimistic about the future? • As much as I ever did (0) • Rather less than I used to (1) • Definitely less than I used to (2) • Nothing (3)		1.59	1.00		1.51	0.91	12.01	0.440
	98 (15.6)			87 (13.3)				
	204 (32.5)			244 (37.2)				
	180 (28.7)			221 (33.7)				
	145 (23.1)			103 (15.7)				
Did you enjoy a good book, the radio or a TV programme? • Often (0) • Sometimes (1) • Not often (2) • Very seldom (3)		1.47	1.17		1.08	0.96	7.07	0.832

	167 (26.6)			202 (30.8)				
	187 (29.8)			272 (41.5)				
	84 (13.4)			102 (15.6)				
	189 (30.1)			79 (12.1)				
Category depression scale: • Normal (≤7) • Risk (8–10) • Depression (≥11)		10.66	5.02		8.87	4.40	4.61	0.832
	182 (29.0)			212 (41.5)				
	134 (21.4)			253 (23.3)				
	311 (49.6)			230 (35.1)				

X: mean; SD: standard deviation; X^2^: Pearson’s chi-square; *p*: *p*-value.

**Table 5 ijerph-18-08310-t005:** HADS results by category.

HADS Results	Nurses (*n* = 431)				ANCTs (*n* = 196)			
	1st Wave (*n*, %)	2nd Wave (*n*, %)	X^2^	*p*	1st Wave (*n*, %)	2nd Wave (*n*, %)	X^2^	*p*
Category anxiety scale: • Normal • Risk • Anxiety			31.91	<0.001 ***			25.81	<0.001 ***
	65 (15.1)	119 (27.6)			20 (10.2)	57 (29.1)		
	87 (20.2)	103 (23.9)			27 (13.8)	36 (18.4)		
	279 (64.7)	209 (48.5)			149 (76.0)	103 (52.5)		
Category depression scale: • Normal • Risk • Depression			10.07	0.018 *			24.35	<0.001 ***
	139 (32.3)	179 (41.5)			32 (16.3)	50 (25.5)		
	98 (22.7)	96 (22.3)			53 (27.0)	79 (40.3)		
	194 (45.0)	156 (36.2)			111 (56.6)	67 (34.2)		

ANCTs: auxiliary nursing care technicians. X^2^: McNemar–Browker test; *p*: *p*-value. *< 0.05; ***< 0.001.

**Table 6 ijerph-18-08310-t006:** Odds ratio anxiety and depression.

	Anxiety					
	1st Wave			2nd Wave		
	OR	CI	*p*	OR	CI	*p*
Gender [Male] • Female						
	0.59	0.35–0.98	0.042 *	0.67	0.47–1.02	0.110
Age [≥56 years] • 16–25 years • 26–35 years • 36–45 years • 46–55 years						
	0.62	0.19–1.99	0.430	1.95	0.67–5.80	0.221
	0.61	0.23–1.60	0.319	1.50	0.61–3.74	0.370
	0.57	0.25–1.32	0.188	1.76	0.80–3.95	0.159
	0.75	0.35–1.61	0.459	1.55	0.74–3.34	0.244
Professional category [Nurse] • ANCTs						
	0.59	0.37–0.92	0.024 *	0.60	0.40–0.88	0.010 **
Work experience [>20 years] • 0–5 years • 6–10 years • 11–20 years						
	1.59	0.74–3.44	0.229	1.30	0.66–2.56	0.439
	1.54	0.75–3.18	0.234	1.26	0.66–2.41	0.467
	0.95	0.53–1.71	0.888	1.06	0.60–1.70	0.939
Isolation by COVID-19 [NO] • YES						
	0.58	0.36–0.90	0.019 *	0.70	0.47–1.02	0.068
A family member was infected COVID-19 [NO] • YES						
	0.69	0.44–1.08	0.115	1.02	0.12–1.46	0.874
Feeling of rejection [NO] • YES						
	0.45	0.29–0.68	<0.001 ***	0.35	0.24–0.51	<0.001 ***
	**Depression**					
	**1st Wave**			**2nd Wave**		
	**OR**	**CI**	***p***	**OR**	**CI**	***p***
Gender [Male] • Female						
	0.37	0.21–0.63	<0.001 ***	0.68	0.39–1.14	0.159
Age [≥56 years] • 16–25 years • 26–35 years • 36–45 years • 46–55 years						
	0.59	0.19–1.83	0.369	1.87	0.60–5.79	0.275
	0.42	0.16–1.06	0.686	1.24	0.50–3.07	0.631
	0.50	0.22–1.10	0.090	1.16	0.52–2.56	0.698
	0.73	0.35–1.52	0.410	1.39	0.65–2.95	0.378
Professional category [Nurse] • TCAE						
	0.55	0.36–0.83	0.004**	0.57	0.38–0.86	0.007 **
Work experience [>20 years] • 0–5 years • 6–10 years • 11–20 years						
	2.46	1.20–5.11	0.014 *	1.63	0.81–3.28	0.168
	2.32	1.17–4.64	0.015 *	1.15	0.59–2.24	0.674
	1.44	0.85–2.47	0.176	1.16	0.69–1.57	0.563
Isolation by COVID-19 [No] • Yes						
	0.97	0.65–1.44	0.894	0.90	0.61–1.35	0.633
A family member was infected COVID-19 [No] • Yes						
	0.58	0.28–0.86	0.008 **	0.71	0.49–1.03	0.074
Feeling of rejection [No] • Yes						
	0.38	0.26–0.55	<0.001 ***	0.33	0.22–0.48	<0.001 ***

OR: odds ratio; CI: confidence interval; *p*: *p*-value. *< 0.05; **< 0.01; ***< 0.001.

## Data Availability

The data are collected in a database prepared by the research team.

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
