# Peer review of "Impact of the COVID-19 Pandemic on the Mental Health of Nurses and Auxiliary Nursing Care Technicians—A Voluntary Online Survey"

_ijerph, 2021, doi:10.3390/ijerph18168310_

Round 1
Reviewer 1 Report
The authors adequately addressed comments and improved the manuscript
Author Response
Dear reviewer:
We appreciate your time in reviewing the manuscript again. Thanks to your comment, the manuscript has been greatly improved.
Kind regards.
Reviewer 2 Report
The author revised the edition in a good manner as concerned some comments. However, there are some important notes to consider.
Some comments:
- check the spaces between words. ex. lines 44 and 46, 107, etc ...
- line 88: delete “In Spain” after the dot.
- Line 158: the data are incorrect. They do not coincide with those reported in table 2.
- Line 161: “ 10.2% took some type of substance…”. During the questionnaire, the authors asked for the use of both pharmacological and natural substances. It is better to specify that the subjects took both of them not “some” substances.
- Line 168: “of the total,…” what is the total? It is not clear. Is it 687 subjects? Is it 680? if they do not specify it, it is not possible to understand what the percentages reported in the same sentence refer to. Furthermore, the calculation of these percentages might seem wrong.
- Line 195: in which table are the data discussed in the sentence? can the authors specify this?
- Check for table 6
the conclusion section is not sufficient.
- Line 260: “.. the score obtained in the HADS scale by other authors are lower…”. According to the authors, what could be the reason for this observed difference?
- Line 303: “protocols, … Like” what it means?
- From the results, we can notice that in the first period the percentages of anxiety and depression were lower in nurses than in ANCTs group. The authors do not discuss these results.
- Furthermore, in the second period the frequencies for anxiety and depression decreased in both groups and there were also significant differences in the two groups in the first and second periods. No one of the results has been discussed by the authors. Can they give some hypotheses on these results?
Author Response
Dear reviewer,
thank you for taking the time to review our manuscript. After reading your comments, the following changes have been made:
Checked the spaces between words throughout the manuscript.
Line 88: deleted "In Spain" after the dot.
Line 158: these data do not appear in the table, as the question was general and not prior to each period. The table shows whether they took substances to relax in either or both periods. And these data show whether respondents took substances to sleep before the pandemic (more general).
Line 161: replaced "some" by "both".
Line 168: this paragraph has been modified.
Line 195: in which table are the data discussed in the sentence? Can the authors specify this?
Checked table 6.
Line 260: added a sentence in this paragraph.
Line 303: this sentence has been amended.
From the results, we can notice that in the first period the percentages of anxiety and depression were lower in nurses than in ANCTs group. The authors do not discuss these results.
The following text appears in the discussion section: “The roles of nurses and ANCTs are different, especially those related to basic patient care, which may influence their exposure time, decreasing or increasing the risk of infection”
Furthermore, in the second period the frequencies for anxiety and depression decreased in both groups and there were also significant differences in the two groups in the first and second periods. No one of the results has been discussed by the authors. Can they give some hypotheses on these results?
The following text was added “This decrease in the second period may be due to the fact that at the beginning of the pandemic, the "first wave", there was a lot of confusion, lack of information, lack of training of these professionals, lack of PPE, high number of infections in both groups and lack of diagnostic tests. In the second period, the training of these professionals improved, as well as the material and human resources for the management of this pandemic”
Again, thank you for your comments which help us to improve our manuscript.
Kind regards.
Reviewer 3 Report
This first para in the results section does not make sense and the figures need amendment. if 655 subjects were included and 680 eliminated for discropancies, and 7 for missing data, then the total must be 655+ 680+7.
"A total of 687 responses were obtained, but 7 were eliminated because of missing data for the variables sex and age. Of these, 680 were eliminated if there were discrepancies between the answers to some questions, for example, not working in a period, but answering affirmatively in the answers related to that period. After eliminating these subjects, the sample consisted of 655 subjects, of whom 627 subjects had worked in the first
wave and 655 subjects in the second wave. The number of subjects who had worked in
both periods was 627"
Otherwise the authors have responded to my concerns
Author Response
Dear reviewer,
thank you for taking the time to review our manuscript. After reading your comment, this paragraph has been modified, as there was an error that we did not see at the time.
Thank you, as your comment has helped us to improve our manuscript.
Kind regards.
Reviewer 4 Report
This ONLINE survey examined the impact of the COVID-19 pandemic on the mental health of nurses and auxiliary nursing care technicians in Spain during 2 waves. Although they contributed the occurrence of anxiety and depression of healthcare workers during 2 waves and compare the difference between two periods, the results were not surprising.
- research significance --I agree that it’s important to study the depression and/or anxiety owing to COVID-19 pandemic among healthcare professionals, which were associated with decreased patient safety and care quality. However, more specific rationale, research significance, or knowledge gap to look at the difference in 2 waves is expected.
- Further, how could you differentiate the natural changes of research variables by time? Rather than merely getting the occurrence, I think, exploring related factors may be more informative and generalizable.
- Instrument validity--I also wonder how did you measure the emotional exhaustion? How did you choose these questions and how to convince that these questions could assess healthcare professionals’ “emotional exhaustion”? Instrument validity from previous studies or theoretical knowledge is needed. What’s difference between these two similarly conceptual variables- “emotional exhaustion” and “emotional distress”?
- Ethical consideration needs to be described in detail.
- Some errors and data in tables mysterious to me. It’s not clear about what statistic methods have been used for some tables. Some content has been repeated in Line 214-215. Table 6, IC or CI?
- Please confirm the research design as cross-sectional design because you measured variables twice.
Author Response
Dear reviewer,
thank you for taking the time to review our manuscript. After reading your comments, the following changes have been made:
- research significance --I agree that it’s important to study the depression and/or anxiety owing to COVID-19 pandemic among healthcare professionals, which were associated with decreased patient safety and care quality. However, more specific rationale, research significance, or knowledge gap to look at the difference in 2 waves is expected.
In the introduction section, the following text has been added “Hence the importance of studying the effect of the pandemic on the mental health of these professionals and how the figures may be modified by time and improved information and training. For this reason, anxiety and depression should be studied in various periods and the changes that have occurred in these periods and how they have affected health professionals.”
- Further, how could you differentiate the natural changes of research variables by time? Rather than merely getting the occurrence, I think, exploring related factors may be more informative and generalizable.
As you comment, the variables studied can undergo natural changes, but the pandemic has been a very stressful situation for all health professionals and perhaps these changes have been more associated with the pandemic or changes in our routines than with natural changes. As this is an anonymous study, we cannot identify the respondents to the survey to look for other related factors, but this comment helps us to improve and plan new projects in this or a similar field. The information that this manuscript intends to give us, is the very high percentages of anxiety and/or depression among nurses and ANCTs, especially associated to factors such as lack of resources, information, training, ...
- Instrument validity--I also wonder how did you measure the emotional exhaustion? How did you choose these questions and how to convince that these questions could assess healthcare professionals’ “emotional exhaustion”? Instrument validity from previous studies or theoretical knowledge is needed. What’s difference between these two similarly conceptual variables- “emotional exhaustion” and “emotional distress”?
To my knowledge the intended use of HADS is for hospitalized patients (focusing on generalized anxiety) rather than employees in healthcare (like nurses and ANCTs).
This method has been used for health professionals in other studies before and after ours:
- Tan BYQ, Kanneganti A, Lim LJH, et al. Burnout and Associated Factors Among Health Care Workers in Singapore During the COVID-19 Pandemic. J Am Med Dir Assoc. 2020;21(12):1751-1758.e5. doi:10.1016/j.jamda.2020.09.035
- Lin J, Ren YH, Gan HJ, Chen Y, Huang YF, You XM. Factors associated with resilience among non-local medical workers sent to Wuhan, China during the COVID-19 outbreak. BMC Psychiatry. 2020;20(1):417. Published 2020 Aug 24. doi:10.1186/s12888-020-02821-8
- Metin N, Turan Ç, Utlu Z. Changes in dermatological complaints among healthcare professionals during the COVID-19 outbreak in Turkey. Acta Dermatovenerol Alp Pannonica Adriat. 2020;29(3):115-122.
- Tasnim R, Sujan MSH, Islam MS, et al. Prevalence and correlates of anxiety and depression in frontline healthcare workers treating people with COVID-19 in Bangladesh. BMC Psychiatry. 2021;21(1):271. Published 2021 May 25. doi:10.1186/s12888-021-03243-w
- Malfa CS, Karaivazoglou K, Assimakopoulos K, Gourzis P, Vantarakis A. Psychological Distress and Health-Related Quality of Life in Public Sector Personnel. Int J Environ Res Public Health. 2021;18(4):1865. Published 2021 Feb 14. doi:10.3390/ijerph18041865.
In addition, this scale was validated for the Spanish adult and healthy population, and not only in hospitalised or ill patients. (Terol, M. C., López-Roig, S., Rodríguez-Marín, J., Martín-Aragón, M., Pastor, M. A., y Reig, M. T. (2007). Propiedades psicométricas de la Escala Hospitalaria de Ansiedad y Estrés (HAD) en población española. Ansiedad y Estrés, 13 (2-3), 163-176.
- Ethical consideration needs to be described in detail.
This section has been modified, as you suggest. The data were anonymised. Anonymisation is a technique that irreversibly alters data so that the data subject is no longer directly or indirectly identifiable. These data are no longer considered personal data (unlike the pseudonymisation technique).
- Some errors and data in tables mysterious to me. It’s not clear about what statistic methods have been used for some tables. Some content has been repeated in Line 214-215. Table 6, IC or CI?
Modified and revised the whole table
- Please confirm the research design as cross-sectional design because you measured variables twice.
In the abstract and material and method it is stated that the study follows a cross-sectional design.
Again, thank you for your comments which help us to improve our manuscript.
Kind regards.
Reviewer 5 Report
Overall
The authors compared the prevalence of depression and anxiety in Spanish nurses and ANCTs during the first and second wave of the SARS-Cov-2 pandemic. They found that depression and anxiety declined „significantly“ between the two phases and that both were „significantly“ less prevalent in ANCTs.
- Due to the non-probability sampling process I would rather report the results descriptively as there might be a greater self selection bias when maximally stressed nurses do not have the time and nerves to participate (making the p-values invalid). Otherwise, I would at least be interested in the distributions.
- To my knowledge the intended use of HADS is for hospitalized patients (focusing on generalized anxiety) rather than employees in healthcare (like nurses and ANCTs).
- Comparing current emotional states to past ones makes the cross-sectional study prone to recall biases which is not reflected critically.
- I would not use the McNemar-Test because many respondents only answered once.
- Numerous formatting and translation errors (s. below) make it rather difficult to read the paper.
- Due to extensive testing of group as well as time-dependent differences I would not speak of a „descriptive“ study (l. 26).
- Perhaps, multiple testing leads to an type 1 error inflation which is not corrected. Moreover odds ratios are interpreted as risk ratios and a multivariate analysis (e.g. regarding sex) is lacking.
- Finally, I would be interested in a report of the pre-Covid prevalences of depression and anxiety in nurses, not least in order to find out why this paper’s findings are not in line with previous research. It seems to be counterintuitive that depression and (generalized) anxiety are decreasing over the course of the pandemic. The role of post-traumatic stress and emotional numbing as an alternative explanation is not addressed.
En detail
- Numerous formatting errors like missing spaces, for example in the following lines:
27 „outusing“, l. 30 „therewere“, l. 39: „thenew“ etc.
- Numerous translation errors like wrong words, for example „nothing“ instead of never (tables 3 and 4) or „y“ instead of and (table 1).
- Some wrong sentence constructions like missing predicates (e.g. ll. 62 and 76 f.).
- I think the correct name is SARS-Cov-2 (l. 40).
- I did not see that „all“ national governments took measures (l. 42) like „total confinement“ (l. 43 f.), before returning to „normality“ (l. 45).
- 46-50: From my point of view, I would not underestimate UV and temperature effects (with people being outside), as SARS-Cov-2 is mostly spread by aerosoles.
- 54 f.: What is this in percentages?
- 65 f.: What about physicians?
- 64: Contrary to insomnia, depression and anxiety, stress is no disorder.
- 130: It can not be true that 680 were eliminated.
- Table 1: If the test is between nurses and ANCTs, there are p-values missing.
- Table 2: I am not sure what are the statistical tests beyond the waves comparison.
- Tables 3 and 4: The categorical limits are unclear (might be due to translation errors).
- 183-185: Sentence doubled.
- 351 f.: Data on mental health is considered pseudo-anonymous rather than anonymous.
Author Response
Dear reviewer,
thank you for taking the time to review our manuscript. After reading your comments, the following changes have been made:
Due to the non-probability sampling process I would rather report the results descriptively as there might be a greater self selection bias when maximally stressed nurses do not have the time and nerves to participate (making the p-values invalid). Otherwise, I would at least be interested in the distributions.
This point has been taken up as limitations in line 320:The generalisability of the results is limited, as it is a non-probability sample and the biases mentioned above. Moreover, as it is a voluntary questionnaire, those nurses and/or ANCTs who are more aware of the problem or who have a greater impact on their emotional state may have participated. However, it should be noted that, although the current research could not be representative of the entire population of nurses and/or ANCTs working in Spain, a broad coverage has been achieved, being a starting point for more specific future research.”
To my knowledge the intended use of HADS is for hospitalized patients (focusing on generalized anxiety) rather than employees in healthcare (like nurses and ANCTs).
This method has been used for health professionals in other studies before and after ours:
- Tan BYQ, Kanneganti A, Lim LJH, et al. Burnout and Associated Factors Among Health Care Workers in Singapore During the COVID-19 Pandemic. J Am Med Dir Assoc. 2020;21(12):1751-1758.e5. doi:10.1016/j.jamda.2020.09.035
- Lin J, Ren YH, Gan HJ, Chen Y, Huang YF, You XM. Factors associated with resilience among non-local medical workers sent to Wuhan, China during the COVID-19 outbreak. BMC Psychiatry. 2020;20(1):417. Published 2020 Aug 24. doi:10.1186/s12888-020-02821-8
- Metin N, Turan Ç, Utlu Z. Changes in dermatological complaints among healthcare professionals during the COVID-19 outbreak in Turkey. Acta Dermatovenerol Alp Pannonica Adriat. 2020;29(3):115-122.
- Tasnim R, Sujan MSH, Islam MS, et al. Prevalence and correlates of anxiety and depression in frontline healthcare workers treating people with COVID-19 in Bangladesh. BMC Psychiatry. 2021;21(1):271. Published 2021 May 25. doi:10.1186/s12888-021-03243-w
- Malfa CS, Karaivazoglou K, Assimakopoulos K, Gourzis P, Vantarakis A. Psychological Distress and Health-Related Quality of Life in Public Sector Personnel. Int J Environ Res Public Health. 2021;18(4):1865. Published 2021 Feb 14. doi:10.3390/ijerph18041865.
In addition, this scale was validated for the Spanish adult and healthy population, and not only in hospitalised or ill patients. (Terol, M. C., López-Roig, S., Rodríguez-Marín, J., Martín-Aragón, M., Pastor, M. A., y Reig, M. T. (2007). Propiedades psicométricas de la Escala Hospitalaria de Ansiedad y Estrés (HAD) en población española. Ansiedad y Estrés, 13 (2-3), 163-176.
Comparing current emotional states to past ones makes the cross-sectional study prone to recall biases which is not reflected critically.
As he comments, in this type of study there are biases when it comes to remembering and interpreting emotional states. This has been added to the manuscript within the limitations.
I would not use the McNemar-Test because many respondents only answered once.
The McNemar-Test was only used on subjects who had worked in both periods, so they had responded twice. When only one period was evaluated, Pearson's Chi-square was used.
Numerous formatting and translation errors (s. below) make it rather difficult to read the paper.
We apologise, it was changed when we changed the formatting. The whole manuscript has been revised again and the problems have been fixed to make it more readable.
Due to extensive testing of group as well as time-dependent differences I would not speak of a „descriptive“ study (l. 26).
Descriptive" has been deleted
Perhaps, multiple testing leads to an type 1 error inflation which is not corrected. Moreover odds ratios are interpreted as risk ratios and a multivariate analysis (e.g. regarding sex) is lacking.
Risk" has been replaced by "probability". No multivariate analysis by gender or occupational category was carried out due to study limitations (written in the manuscript), but we take note, for our next project.
Finally, I would be interested in a report of the pre-Covid prevalences of depression and anxiety in nurses, not least in order to find out why this paper’s findings are not in line with previous research. It seems to be counterintuitive that depression and (generalized) anxiety are decreasing over the course of the pandemic. The role of post-traumatic stress and emotional numbing as an alternative explanation is not addressed.
Anxiety and depression were higher at the beginning due to the lack of information and training in the management of this pandemic. The lack of PPE, the lack of training to put it on safely, and the number of infections in nurses and ANCTs, made both groups more anxious and depressed. In the second wave, there was more information, better protocols, more resources, ..., which would help to reduce anxiety and depression. I speak from my experience as a nursing supervisor, coordinating 32 nurses and 24 ANCTs.
.
En detail
Numerous formatting errors like missing spaces, for example in the following lines:
27 „outusing“, l. 30 „therewere“, l. 39: „thenew“ etc.
All formatting and spelling errors have been checked and corrected.
Numerous translation errors like wrong words, for example „nothing“ instead of never (tables 3 and 4) or „y“ instead of and (table 1).
For some questions the answer was "nothing" and for others "never".
I think the correct name is SARS-Cov-2 (l. 40).
Modified.
I did not see that „all“ national governments took measures (l. 42) like „total confinement“ (l. 43 f.), before returning to „normality“ (l. 45).
All national governments took some kind of measures, which may or may not be total confinement. For example: use of masks, not allowing foreigners to enter the country, ...
46-50: From my point of view, I would not underestimate UV and temperature effects (with people being outside), as SARS-Cov-2 is mostly spread by aerosoles.
We have not found evidence of a relationship with UV and temperature, although this is an interesting topic. It was thought that external temperature influenced the incidence of COVID-19, with a higher incidence in winter than in summer. But this data is being contradicted by the "fifth wave" in Spain.
54 f.: What is this in percentages?
The number of infected health professionals out of the total population or the number of infected.
65 f.: What about physicians?
As stated in that paragraph, nurses and ANCTs are the largest group in healthcare practice in Spain and all of their work is carried out in close contact with patients. Moreover, the techniques performed on a regular basis require closer contact with patients. For this reason, and due to the large number of infected nurses and ANCTs, it was decided to carry out the study on these groups.
64: Contrary to insomnia, depression and anxiety, stress is no disorder.
It is indeed not a disorder, but the physical and psychological burden on nurses and ANCTs can increase stress.
130: It can not be true that 680 were eliminated.
This is a mistake that has been changed.
Table 1: If the test is between nurses and ANCTs, there are p-values missing.
Revised and modified the table. It was a transcription error.
Table 2: I am not sure what are the statistical tests beyond the waves comparison.
In table 2, Pearson's Chi-square was performed.
Tables 3 and 4: The categorical limits are unclear (might be due to translation errors).
We have not found the errors you are referring to. If you can guide us, we would appreciate it.
183-185: Sentence doubled.
Modified
351 f.: Data on mental health is considered pseudo-anonymous rather than anonymous.
Pseudonymisation is a process that allows you to change the original data set (e.g. email) to an alias or pseudonym. Pseudonymisation is a reversible process, which encrypts the data but allows for re-identification later if necessary.
Anonymisation is a technique that irreversibly alters data so that the data subject is no longer directly or indirectly identifiable. These data are no longer considered personal data (unlike the pseudonymisation technique).
The latter technique is the one used in our study.
Again, thank you for your comments which help us to improve our manuscript.
Kind regards.
Round 2
Reviewer 5 Report
Dear authors,
thank you for the friendly correspondence.
Although I am convinced that there were learning effects affecting the mental health of the respondents and the paper shows already some decent improvements, it still cannot be published in the present form (from my point of view). This is mainly due to the fact that the assumptions for statistical testing, i.e. an independent, identically distribution of the subject’s probability of participating, are violated. Instead, I would recommend stating that it is an explorative study with descriptive analysis but without any significance testing (p-values etc.) used. This would also eliminate the need for a correction of multiple testing and multivariate instead of naïve analysis.
Besides recall bias, there are potential confounders of the longitudinal comparison that are not addressed, like the weather (affecting both covid- and depression prevalence) and the development/use of partially effective medicine (e.g. monoclonal antibodies, corticoids) as well as virus variants affecting lethality.
Your statement regarding the validity of the HADS for the general population is right and also backed up by https://pubmed.ncbi.nlm.nih.gov/11832252/. There might be confusion due to some country-specific versions are validated for patients only. This could be cleared up by citing some of the papers you mentioned (e.g. the latter).
“Risk has been replaced by ‘probability’”, but has to be replaced by chance or odds.
“Anxiety and depression were higher at the beginning due to the lack of information and training in the management of this pandemic.” The formulated mono-causal relation is not backed up by the data in this markedness and the decline still seems counterintuitive to me because depression and generalized (sic) anxiety take time to develop (while more family members were infected within the second wave). The role of emotional numbing as a consequence of post-traumatic stress and an alternative explanation for the decline is still not addressed. “I speak from my experience as a nursing supervisor, coordinating 32 nurses and 24 ANCTs.” – Unfortunately, this is (so-far) unpublished anecdotal evidence.
“All formatting and spelling errors have been checked and corrected.” – This is not true as there are still errors beginning with the title (“on line” instead of online), which might be due to the formatting.
“For some questions the answer was ‘nothing’ and for others ‘never’" – Besides translation and tempus errors (table 4, first question), “nothing” is still the wrong translation.
“All national governments took some kind of measures, which may or may not be total confinement. For example: use of masks, not allowing foreigners to enter the country.” – There were countries (e.g. Turkmenistan) without any state (vs. civil) protective measures for a long time and total confinement sounds to me as if there were continuous bans on going out. Although this is negligible. J
“We have not found evidence of a relationship with UV and temperature, although this is an interesting topic. It was thought that external temperature influenced the incidence of COVID-19, with a higher incidence in winter than in summer. But this data is being contradicted by the "fifth wave" in Spain.”
For evidence on temperature and UV see e.g. https://www.pnas.org/content/117/44/27456.short. A fifth wave in Spain does not contradict the data because it might be a relative reduction and we do not know how this fifth wave would have been like in Winter.
“The number of infected health professionals out of the total population or the number of infected.” – I meant the percentages for France, Italy and Germany.
“We have not found the errors you are referring to. If you can guide us, we would appreciate it.” – E.g. I refer to the gradation/difference between “not very often” and “rarely” or “occasionally” and “sometimes” which might be traced back to translation errors.
“In table 2, Pearson's Chi-square was performed.” – I wonder, (beyond the waves comparison on the right) what is tested/what are the hypotheses of the Chi-square tests in columns 2 and 4 (p-values).
Finally, I am not sure whether it is “emotional exhaustion”, which is measured (validly) in the second part.
Nevertheless kindest regards and I hope a (completely revised) version will be published somewhere, because studying the mental health of nurses and ANCTs is of high importance!
Author Response
Dear reviewer,
Once again we thank you for your comments, and for the time you are taking to review our manuscript.
In response to your suggestions, changes have been made to the manuscript, with the aim of meeting your expectations.
The responses to your comments are added below.
Besides recall bias, there are potential confounders of the longitudinal comparison that are not addressed, like the weather (affecting both covid- and depression prevalence) and the development/use of partially effective medicine (e.g. monoclonal antibodies, corticoids) as well as virus variants affecting lethality.
This point has been added in the limitations.
Your statement regarding the validity of the HADS for the general population is right and also backed up by https://pubmed.ncbi.nlm.nih.gov/11832252/. There might be confusion due to some country-specific versions are validated for patients only. This could be cleared up by citing some of the papers you mentioned (e.g. the latter).
A sentence has been added to this section.
“Risk has been replaced by ‘probability’”, but has to be replaced by chance or odds.
In the previous amendments, risk was replaced by probabilities.
“Anxiety and depression were higher at the beginning due to the lack of information and training in the management of this pandemic.” The formulated mono-causal relation is not backed up by the data in this markedness and the decline still seems counterintuitive to me because depression and generalized (sic) anxiety take time to develop (while more family members were infected within the second wave). The role of emotional numbing as a consequence of post-traumatic stress and an alternative explanation for the decline is still not addressed. “I speak from my experience as a nursing supervisor, coordinating 32 nurses and 24 ANCTs.” – Unfortunately, this is (so-far) unpublished anecdotal evidence.
The data obtained in the present study show lower figures for anxiety and depression in the second wave, showing that lack of information, training and lack of resources increased anxiety and depression. In the second wave, the increase of material and human resources (oxygen therapy, PPE, ...) improved the mental health of the surveyed subjects.
We only intended to contextualise the response to your comment. We understand that it is not evidence but it was a starting point for us to consider this project.
“All formatting and spelling errors have been checked and corrected.” – This is not true as there are still errors beginning with the title (“on line” instead of online), which might be due to the formatting.
The manuscript has again been revised by the authors and by an official translator to correct spelling mistakes. The error you mention, online or on line, can be spelled either way, but it is more correct to use the term online.
“For some questions the answer was ‘nothing’ and for others ‘never’" – Besides translation and tempus errors (table 4, first question), “nothing” is still the wrong translation.
Tables 3 and 4 have been amended.
“All national governments took some kind of measures, which may or may not be total confinement. For example: use of masks, not allowing foreigners to enter the country.” – There were countries (e.g. Turkmenistan) without any state (vs. civil) protective measures for a long time and total confinement sounds to me as if there were continuous bans on going out. Although this is negligible.
Replaced "all" by "most".
“We have not found evidence of a relationship with UV and temperature, although this is an interesting topic. It was thought that external temperature influenced the incidence of COVID-19, with a higher incidence in winter than in summer. But this data is being contradicted by the "fifth wave" in Spain.” For evidence on temperature and UV see e.g. https://www.pnas.org/content/117/44/27456.short. A fifth wave in Spain does not contradict the data because it might be a relative reduction and we do not know how this fifth wave would have been like in Winter.
After carefully reading the proposed paper ( https://www.pnas.org/content/117/44/27456.short), the association between climate and transmission is not yet very clear, although they predict that COVID-19 infection will temporarily decline during the summer, recover in autumn and peak next winter. However, uncertainty remains high and many factors in addition to climate, such as social interventions, will influence transmission. Many factors influence the increase in transmission, especially social contacts.
A new paragraph on possible biases related to context and time of year is added in limitations.
“The number of infected health professionals out of the total population or the number of infected.” – I meant the percentages for France, Italy and Germany.
In the reference where these figures have been obtained, only the total numbers appear and not the percentages. I have made a new search but have not found any reference to these percentages.
“We have not found the errors you are referring to. If you can guide us, we would appreciate it.” – E.g. I refer to the gradation/difference between “not very often” and “rarely” or “occasionally” and “sometimes” which might be traced back to translation errors.
The tables have been rechecked and errors have been corrected.
“In table 2, Pearson's Chi-square was performed.” – I wonder, (beyond the waves comparison on the right) what is tested/what are the hypotheses of the Chi-square tests in columns 2 and 4 (p-values).
The observed frequencies are tested to ensure that they are different from the expected frequencies for each category of each variable. That is, that they follow a normal distribution and are the same for all of them.
Finally, I am not sure whether it is “emotional exhaustion”, which is measured (validly) in the second part.
The project only aims to measure the presence of anxiety and depression and not emotional exhaustion, which can be a risk factor. These results help to improve the supply of mental health services for these professionals (this point is added in conclusions).
Again, thank you for your comments which help us to improve our manuscript.
Kind regards.
This manuscript is a resubmission of an earlier submission. The following is a list of the peer review reports and author responses from that submission.
Round 1
Reviewer 1 Report
The authors present findings on the mental health of nurses and auxiliary nursing care technicians in Spain. This paper contributes to the literature by adding evidence on Spain healthcare workers' mental health condition during Covid-19. The paper is easy to follow but it needs some minor editing.
Content comments
- The authors should abstain to use the term “prevalence” of depression and anxiety” in the paper as the HADS detects mental disorder symptoms that not necessarily translate into diagnosis/ prevalent cases. Anxiety and depressive symptoms should be used, and if the cut off its specific and it uses the highest cut off then the authors may use the term probable depression and anxiety
- In the description of the instruments, authors should briefly explain how the instrument work, and more importantly, they should specify what cut-off was used to categorise people into the anxiety and depressive symptoms pool.
- In the discussion, the authors should acknowledge that they used convenience sampling and explicitly state its implications (e.g., results cannot be generalised to all nurses, results are likely to be overestimated as people that respond to social media adverts may experience more anxiety and depression than those not engaged in social media)
esthetic comments
- The indentation of the tables is all over the place (e.g., table 1 male and female)
- Line 60- use “among others” instead of “… “
- There are a couple of spaces in between words
Author Response
Dear reviewer,
Firstly, we appreciate the time dedicated to our manuscript, as well as the clarifications you request, which help us to understand the doubts that a future reader may have, if the manuscript gets published.
Secondly, we answer to the questions that you have made, with aim of resolving doubts raised by our manuscript.
The term "prevalence" has been replaced by "symptoms".
On line 101 begins the paragraph describing the instrument used, as well as the cut-off points “The third section measures the degree of emotional distress (anxiety and depression) using the Hospital Anxiety and Depression Scale (HADS) [24] validated for the Spanish population [25]. The HADS is an instrument widely used to assess symptoms of anxiety and depression in the hospital setting. It includes 14 items divided into two subscales (anxiety and depression), each with 7 items. The items are scored on a Likert-type scale (0, 1, 2 and 3). The total scale score ranges from 0 to 21. The cut-off points for the different categories in the two subscales are: normal score (≤ 7), risk (8-10) and anxiety/depression (≥ 11).
A paragraph on limitations has been added: “The retrospective nature may lead to limitations in the study, as respondents sometimes do not remember everything that happened in a specific period, although the severity of this pandemic makes us not forget the moments experienced and remember them accurately. In addition, the COVID-19 pandemic has increased the care burden of nurses and ANCTs, and therefore, it was not possible to carry out the questionnaire during critical periods. The aim was to find out if there was any difference between the 1st and 2nd wave, as protocols, resources and lack of knowledge were not similar in these two periods. The use of online questionnaires has increased exponentially during the pandemic, due to restrictions due to the COVID-19 pandemic. This may result in under-representation of the entire study population (people without access). Therefore, a mass dissemination was carried out with the help of professional associations, associations, scientific societies, among others”.
All tables have been revised and the identification has been modified.
As suggested by you, "..." has been changed to "among others".
Once again, we appreciate the time and attention dedicated to our manuscript. We really hope we have reached your expectations, with the modifications made and that the explanations to those that we have not modified be considered as appropriate.
Kind regards.
Reviewer 2 Report
1. please, check the spaces between words throughout the text 2. line 41. Check for the reference. It does not seem suitable for the sentence. 3. Line 63. ANCT: it is more correct to put the definition of the acronym the first time it is mentioned in the text. “auxiliary nursing care technicians (ANCTs)”. 4. Line 109. “Paragraphs 2 and 3 are duplicated …”: it is better to use the word “section”, not “paragraphs”. 5. Line 111 and following: the sections of the questionnaire that had been duplicated ("1st wave" and for the "2nd wave") were administered at the same time? 6. Line 124. I don’t know the “SPPS-25 software”. Is it possible that the authors refer to the SPSS -25 software? Please, check for this. 7. Line 131. In Table 1. Socio-demographic and occupational variables. What are the p-values referred to? the authors do not discuss these p-values in the text. 8. Line 142. The value of 23.2% is not correct. It does not correspond to the value shown in the table 9. Line 143. “There were statistically significant differences between the 1st and 2nd waves (p < 0.001)”. Were these differences present in all the factors considered? Specify this. 10. Table 2. Please, align the data in the table. It is not easy to consult the table. 11. Line 149 and following. It is better to express in the table the data “before the pandemic”. 12. Line 151. “When asked whether they….”. From the sentence, it is understood that only the p-value present in the text refers to the use of natural substances compared to pharmacological ones to facilitate sleep. In the table, there are other p-values. they are not covered by the authors. 13. Line 158 and line 184. In the text, it is not clear how many subjects worked in the two periods under study and how many only in one period. the sentences reported do not coincide. 14. Line 198. “There were significant differences in the two groups in the first and second periods (Table 5).”. were there also differences between the two groups? Did the authors analyze the differences between the groups? 15. Line 242. What area? 16. Line 252. according to the authors, why are their results greater than those in the literature? 17. Line 257. “Our results show….” How do the authors explain Could these results be due to a higher percentage of women in the ANCTs group? the differences between women and men and between ANCTs and nurses? 18. Line 274. “Our results..”: specify. what are the results the authors refer to? 19. Line 282. “HCWs”, what it means? 20. the conclusions are little exhaustiveAuthor Response
Dear reviewer,
Firstly, we appreciate the time dedicated to our manuscript, as well as the clarifications you request, which help us to understand the doubts that a future reader may have, if the manuscript gets published.
Secondly, we answer to the questions that you have made, with aim of resolving doubts raised by our manuscript.
The proposed amendments have been implemented. Below, we respond to those clarifications that we believe have generated the most doubts.
- Line 111 and following: the sections of the questionnaire that had been duplicated ("1st wave" and for the "2nd wave") were administered at the same time?
They were collected at the same time, as the questionnaire was carried out at the same time.
- Line 131. In Table 1. Socio-demographic and occupational variables. What are the p-values referred to? the authors do not discuss these p-values in the text.
The p-values refer to the differences between the group of nurses and ANCTs. They are not discussed in the text because we believe their use in the table is very graphic.
- Line 143. “There were statistically significant differences between the 1st and 2nd waves (p < 0.001)”. Were these differences present in all the factors considered? Specify this.
It referred to the previous point. It has been amended.
- Table 2. Please, align the data in the table. It is not easy to consult the table.
All tables have been revised and the identification has been modified.
- Line 158 and line 184. In the text, it is not clear how many subjects worked in the two periods under study and how many only in one period. the sentences reported do not coincide.
As shown in line 158, 627 subjects worked during the first period and 655 during the second period. Line 184 shows that 627 subjects worked in both periods, i.e. the number of subjects who did not work in the first period and worked in the second period was 28 (655-627=28).
In the section where line 158 appears we study the number of subjects separately and in the section of line 184 those who worked in both periods, in order to establish the differences or similarities obtained in both periods by the same subjects.
- Line 198. “There were significant differences in the two groups in the first and second periods (Table 5).”. were there also differences between the two groups?
The differences between the two groups were not studied, only in the responses obtained by the members of each group.
- Line 252. according to the authors, why are their results greater than those in the literature?
It may be due to the sample under study, the characteristics of the population, social differences or perception of fear among others.
- Line 257. “Our results show….” How do the authors explain Could these results be due to a higher percentage of women in the ANCTs group? the differences between women and men and between ANCTs and nurses?
The presence of women is higher in these professions, and this may explain the results obtained. The roles of nurses and ANCTs are different, especially those related to basic patient care, which may influence their exposure time, decreasing or increasing the risk of infection. Added to the manuscript
- Line 274. “Our results..”: specify. what are the results the authors refer to?
They appear between " "
- the conclusions are little exhaustive
The conclusions have been amended.
Once again, we appreciate the time and attention dedicated to our manuscript. We really hope we have reached your expectations, with the modifications made and that the explanations to those that we have not modified be considered as appropriate.
Kind regards.
Reviewer 3 Report
This is a timely study of an important topic. Unfortunately its value is severely limited by the research design which is a voluntary on line survey, with no specific sampling methodology.
I think this should therefore be indicated in the title of the paper eg "An exploratory study of the impact.....technicians-a voluntary on line survey".
Such volunteer samples are subject to many sources of bias, and so are only good for generating hypotheses, but not for drawing conclusions, especially as there is no control group who were not subject to the pandemic or lockdowns.
Page 2 Materials and Methods, Study design....the authors need to make clear here that this is a voluntary on line sample...perhaps give some of the wording of the invitation to participate.
How long did it take to accumulate the sample, and how did the timings correlate with the onset of the epidemic in Spain and the various lockdowns in Spain.
Were the initial 687 volunteers used for the 2nd wave of questionnaires, or was the 2nd wave an entirely different group, or a mixture. ..ie was the 2nd wave a follow up study of the first wave group, and what was the time lag?
Sample size calculation...I am not sure how far this applies to volunteer samples as opposed to random samples. Can the authors say something about this.
Section 2.3 implies that data was only collected in the 3rd wave...ie that answers were retrospective for 1st and 2nd wave . I think the reader would benefit from a diagram indicating the timing of the pandemic, and its waves, the timing of data collection , and the periods it covered....there seems to be some confusion in the text about this that needs clarification.
Results
687 responses obtained..did this apply to all the waves or was there some loss?
Discussion-when discussing other studies, attention needs to be paid to their methodology, as you can only interpret and compare results in this light .
Limitations
- the volunteer nature of the sample is the most important limitation which means that there will be bias in the group who responded (and most likely be those who were more rather than less affected by the pandemic).
- If I am right in understanding that data was only collected at the 3rd wave, and was therefore retrospective in nature, then this is a major limitation as we know people do not have good recollection of symptoms more than a week or so ago. This is why cross sectional epidemiological studies focus on symptoms in the past week.
Author Response
Dear reviewer,
Firstly, we appreciate the time dedicated to our manuscript, as well as the clarifications you request, which help us to understand the doubts that a future reader may have, if the manuscript gets published.
Secondly, we answer to the questions that you have made, with aim of resolving doubts raised by our manuscript.
The proposed amendments have been implemented. Below, we respond to those clarifications that we believe have generated the most doubts.
This is a timely study of an important topic. Unfortunately its value is severely limited by the research design which is a voluntary on line survey, with no specific sampling methodology.
During the pandemic, due to restrictions, the use of these online questionnaires and this type of methodology has been used in most research of this type or other studies (food consumption, physical activity among others, during the pandemic). A paragraph on limitations related to this point has been added.
I think this should therefore be indicated in the title of the paper eg "An exploratory study of the impact.....technicians-a voluntary on line survey".
The authors believe that it is not necessary to add "a voluntary on line survey", as this point is reflected in the methodology. After reviewing the scientific literature on the use of this type of methodology, we have noted that it does not appear in the title.
Such volunteer samples are subject to many sources of bias, and so are only good for generating hypotheses, but not for drawing conclusions, especially as there is no control group who were not subject to the pandemic or lockdowns.
The aim is to study how the pandemic may affect the mental health of those involved in the provision of health care. We do not intend to compare these results with those obtained with other professionals or in other periods outside the pandemic or confinement. We believe that the existence of a control group is not necessary for this type of study, as we only intend to describe the appearance of symptoms of anxiety and/or depression in nurses and ANCTs during the different periods of the pandemic, especially those considered to be more stressful due to the increase in infections.
We are aware that there may be biases due to the methodology used, which is why we reflect this in the limitations. We have tried to resolve some biases present in other research, but due to the current situation it has been difficult to resolve all biases.
Page 2 Materials and Methods, Study design....the authors need to make clear here that this is a voluntary on line sample...perhaps give some of the wording of the invitation to participate.
A sentence has been added to the methodology and appears more clearly and concisely in section 2.3 Data collection.
How long did it take to accumulate the sample, and how did the timings correlate with the onset of the epidemic in Spain and the various lockdowns in Spain.
In section 2.3 Data collection, the data collection times are shown. In the objective, the months comprising the two periods are shown, as set out by the national health authorities. "The aim of this study is to determine the symptoms of depression and/or anxiety among nurses and ANCTs during the periods known as "1st wave" (March-June) and "2nd wave" (September-November) of the COVID-19 pandemic in Spain, as well as the differences between both periods".
Were the initial 687 volunteers used for the 2nd wave of questionnaires, or was the 2nd wave an entirely different group, or a mixture. ..ie was the 2nd wave a follow up study of the first wave group, and what was the time lag?
As shown in section 2.3 Data collection, all 687 subjects responded to the survey in the same period, but among them there were subjects who worked in both periods or in one of them. Therefore, the results section reflects the data separately (reflecting the occurrence of anxiety and depressive symptoms by period) and for those subjects who worked in both periods (n=627) in order to make a comparison between the two periods.
Sample size calculation...I am not sure how far this applies to volunteer samples as opposed to random samples. Can the authors say something about this.
This point was added because in articles published with the same methodology, but in a different field, we were asked to calculate the sample size. We thought it was not necessary, but we have adapted to what was requested by other journals from the same publisher.
Section 2.3 implies that data was only collected in the 3rd wave...ie that answers were retrospective for 1st and 2nd wave . I think the reader would benefit from a diagram indicating the timing of the pandemic, and its waves, the timing of data collection , and the periods it covered....there seems to be some confusion in the text about this that needs clarification.
The term "3rd wave" has been removed from this section and only the time interval of data collection has been left, in order to avoid possible confusion for future readers, should the results be published.
Results
687 responses obtained..did this apply to all the waves or was there some loss?
A clarifying paragraph has been added in the results section
Discussion-when discussing other studies, attention needs to be paid to their methodology, as you can only interpret and compare results in this light.
The discussion has been reviewed
Limitations
- the volunteer nature of the sample is the most important limitation which means that there will be bias in the group who responded (and most likely be those who were more rather than less affected by the pandemic).
- If I am right in understanding that data was only collected at the 3rd wave, and was therefore retrospective in nature, then this is a major limitation as we know people do not have good recollection of symptoms more than a week or so ago. This is why cross sectional epidemiological studies focus on symptoms in the past week.
A paragraph has been added on the limitations of the study, especially those related to the retrospective nature and the use of online questionnaires.
Once again, we appreciate the time and attention dedicated to our manuscript. We really hope we have reached your expectations, with the modifications made and that the explanations to those that we have not modified be considered as appropriate.
Kind regards.
Reviewer 4 Report
Congratulations on the work, I would like to know if you have counted the possibility of the other questionnaires being more general and not only used in the hospital setting. I have not found a categorization by services that is extremely important in what the covid plants have suffered under pressure, in relation to other services. Service ratios and other factors that have modulated the response of health sciences workers to COVID. The results tables should be clearer and not lose interesting data between data that are not, rewrite them.Author Response
Dear reviewer,
Firstly, we appreciate the time dedicated to our manuscript, as well as the clarifications you request, which help us to understand the doubts that a future reader may have, if the manuscript gets published.
Secondly, we answer to the questions that you have made, with aim of resolving doubts raised by our manuscript.
A categorisation by services was made, but during the piloting of the questionnaire in 20 volunteers, it had to be eliminated because they responded that at the beginning of each wave their unit might not hospitalised patients with COVID-19 infection, but as the days went by they had to do so. In other words, resources were very changeable, as most health centres improvised because it was a very new situation and it was not known how it would evolve. In addition, the ratios were very changeable, as many nurses and ANCTs were isolated due to COVID-19 infection or direct contact, and sometimes there were not enough human resources to cover these absences.
Our experience has shown that patients, supposedly without COVID-19 infection, started with symptomatology and after performing PCR, their results were positive.
All the tables have been reviewed.
Once again, we appreciate the time and attention dedicated to our manuscript. We really hope we have reached your expectations, with the modifications made and that the explanations to those that we have not modified be considered as appropriate.
Kind regards.
Round 2
Reviewer 2 Report
Dear authors,
thanks for your answers to my questions, some doubts have been resolved.
best regards
Reviewer 3 Report
I am afraid I still consider it necessary for the authors to insert the fact that this is a voluntary online sample into the article title. The prevalence rates found in this study are meaningless as they are subject to so many unquantified biases, and there is no sample denominator . The fact that other articles have been published using similar methodology I am afraid is no defence. I remain unhappy about the publication of this article. The fact of the pandemic does not make it impossible to have a defined sample frame and sample, and many epidemiologists are conducting proper surveys during this period.
Reviewer 4 Report
After reviewing the reports of the rest of the reviewers, and reassessing the work, I find that the requested changes have not been carried out in most of the assignments. The article does not vary substantially from the one previously presented.